# Spectrum Index for Estimating Ground Water Content Using Hyperspectral Information

Kicheol Lee [1], Ki Sung Kim [1], Jeongjun Park [2,*] and Gigwon Hong [3,*]

1    Corporate Affiliated Research Institute, UCI Tech, 313, Inha-ro, Michuhol-gu, Incheon 22012, Korea
2    Incheon Disaster Prevention Research Center, Incheon National University, 119 Academy-ro, Yeonsu-gu, Incheon 22012, Korea
3    Department of Civil Engineering, Halla University, 28 Halladae-gil, Wonju-si 26404, Korea
*    Correspondence: smearjun@hanmail.net (J.P.); g.hong@halla.ac.kr (G.H.); Tel.: +82-10-4722-2971 (J.P.); +82-10-9015-6263 (G.H.)

**Abstract:** Quality control considerably affects road stability and operability and is directly linked to the underlying ground compaction. The degree of compaction is largely determined by water content, which is typically measured at the actual construction site. However, conventional methods for measuring water content do not capture entire construction sites efficiently. Therefore, this study aimed to apply remote sensing of hyperspectral information to efficiently measure the groundwater content of large areas. A water content prediction equation was developed through an indoor experiment. The experimental samples comprised 0–40% (10% increase) of fine contents added to standard sand. As high water content is not required in road construction, 0–15% (1% increase) of water content was added. The test results were normalized, the internal and external environments were controlled for precise results, and a wavelength–reflection curve was derived for each test case. Data variability analyses were performed, and the appropriate wavelength for water content reflection, as well as reflectance, was determined and converted into a spectrum index. Finally, various fitting models were applied to the corresponding spectrum index for water content prediction. Reliable results were obtained with the reflectance corresponding to a wavelength of 720 nm applied as the spectrum index.

**Keywords:** spectrum index; water content; hyperspectral information

## 1. Introduction

The term sustainability refers to the ability to maintain a function and survive over time [1]. Several other definitions of the term have also been provided. For example, Brown [2] defined sustainability as the ability to meet the needs of future generations without reducing their opportunities, and the Brundtland Commission [3] defined it as technology to meet present needs without compromising the resources of future generations. In the field of engineering, especially geotechnical engineering, sustainability refers to the use of resources at a low cost while appropriately controlling harmful emissions. The term is divided into four concepts [4,5]: (1) robust design and construction, including social cost and inconvenience caused by construction; (2) the minimized use of resources and energy in the planning, design, construction, and maintenance of geotechnical facilities; (3) the use of methods and materials with a low impact on ecology and the environment; and (4) the reuse of geotechnical structures for minimizing waste.

The concepts of sustainability have been applied to the field of road construction, which is a subfield of geotechnical engineering. Corriere and Rizzo [6] defined "sustainable roads" as roads that can achieve basic design goals (compliance, safety, ease of mobility, maintenance, energy efficiency, transport capacity, etc.) during the construction, maintenance, and operation phases. Meanwhile, the sustainability of roads is mainly considered

from the perspective of maintenance. Greenroads Foundation (USA) evaluated road sustainability using scores obtained from four-year tests on more than 50 types of roads The London Councils in the UK manages a "Highways Minor Works" toolkit to support the procurement of highway services, such as recycling, reducing transport distances, reducing overall lifetime costs, energy use, and $CO_2$ emissions [7,8]. In South Korea, development and research on smart roads are being actively conducted, which mainly requires automation equipment, datafication of information, and accurate quality control in a wide area [9,10].

In addition to the maintenance aspect, the practical implementation of "sustainable roads" is achieved through thorough, early-stage road excavation and pavement work. The compaction performed to secure the road bearing capacity not only ensures the durability of the asphalt or concrete road, but is also the determining process for road performance and quality, including the drivability of vehicles [11]. In general, the goal of compaction is 90% or more, which is calculated as the ratio of the dry unit weight on-site to the maximum dry unit weight. Furthermore, the water content used for the calculation is the ratio of soil water to soil weight, which affects the long-term stability of the subgrade, the quality of compaction, and the number of passes. It is essential to determine the groundwater content prior to performing actual compaction [12–20].

Groundwater content measurement is performed by the traditional methods of comparing the weight of an on-site sample with the weight of the sample after drying, using a scale; time domain reflectometry through the reverberation time of an electrical signal; and ground penetrating radar (GPR) through the intermittent measurement of water volume and dielectric constant [14–19]. However, the disadvantage of these methods is that they are time-consuming and labor-intensive in determining the water content distributed over the entire construction site by point measurement. Furthermore, the passive aspect of the data measurement process may vary the measured values depending on the skill level of the operator or increase the error range, which may cause reliability problems.

Accordingly, it is necessary to measure the water content in a range by a more reliable method rather than by the existing point measurement method. For example, remote hyperspectral sensing has been actively researched recently as a viable method. The field of remote hyperspectral sensing is broadly divided into spectroscopy, radiative transfer, imaging spectroscopy, and hyperspectral image processing, where spectral curves are derived through radiative transfer. A spectrum is a function of wavelength and indicates the distribution of reflectance; thus, the reflectance shown by the spectrum depends on the characteristics of the object [20]. In the construction industry, remote hyperspectral sensing usually involves the use of drones. Through this, orthographic images and hyperspectral information of a wide area are acquired and mainly used to classify mineral types, sizes, and qualities or to analyze vegetation distribution [21–27]. The photographed hyperspectral information indicates only the reflectance based on wavelength and is expressed as a spectrum index by substituting the equation corresponding to each property. The spectrum index is a value obtained by converting the spectral information (wavelength–reflection curve) obtained through a spectral experiment into a single value; this is equivalent to normalizing the necessary information. Thus, secondary processing of hyperspectral information is required to convert the photographed copy to suit the operator's intention.

Measuring groundwater content in ranges requires a conversion of the measured hyperspectral information into a spectrum index representing the water content. However, studies on the spectrum index related to water content have mainly focused on factors influencing water quality or moisture content in [25–29]. Through hyperspectral image analysis, Prošek et al. [28] classified the local waters, and Guo et al. [29] analyzed only the color change of the lake. On the other side, Ge et al. [30] verified the equation for calculating the spectrum index of various water contents to confirm soil aggregate structure and nutrient status. However, in general, many spectrum index models have a small $R^2$ value of calculated water content and measured water content. The most suitable

spectrum index model still has a disadvantage: the target site does not reflect the low water content of agricultural land (measured water content of 10–30%).

The ground targeted in this study was a road construction site. In such a ground, compaction is usually carried out after filling, and the transported soil usually has a similar water content. That is, the water content measured at the surface represents the water content of the entire ground. In this ground, as work was performed on general sand, high water content was not recorded, but low water content was considered. Therefore, the application of the corresponding spectrum index model may show inappropriate results, necessitating the development of a spectrum index to indicate new water content. Thus, ground hyperspectral information was acquired from water content through a normalized and thoroughly controlled indoor experiment in this study. A subsequent spectrum index expressing water content through various combinations was obtained. The soil used in the experiment was prepared from 0 to 30% of fine particles (10% increments; particle size $\leq 0.075$ mm) using standard sand, and it was used to simulate various types of soil while increasing water content by 1%. As the spectrum index must be used during image acquisition through drones, an indoor experimental system with the same measurement method was created, in which hyperspectral information was acquired for spectrum index calculation.

## 2. Methodology for Estimating Ground Water Content in Road Construction Site

The water content measurement process of this study is illustrated in Figure 1 and detailed as follows; (1) A hyperspectral camera is mounted on a drone, which is an unmanned aerial vehicle, and orthographic image and hyperspectral information of the ground for which water content is to be determined are acquired. In this case, the hyperspectral information is the pixel unit of the measurement area, which can be freely adjusted; (2) The acquired image is corrected through post-processing because there is shaking during the drone shooting process using the push-broom method. (3) Hyperspectral information existing in each pixel is converted into water content for a photographed copy where all processes have been completed. Here, hyperspectral information is the relationship between reflectance and wavelength; (4) A specific color is assigned to the water content converted for each pixel, and this is displayed on the map. That is, it implements a color-coded map (CCM).

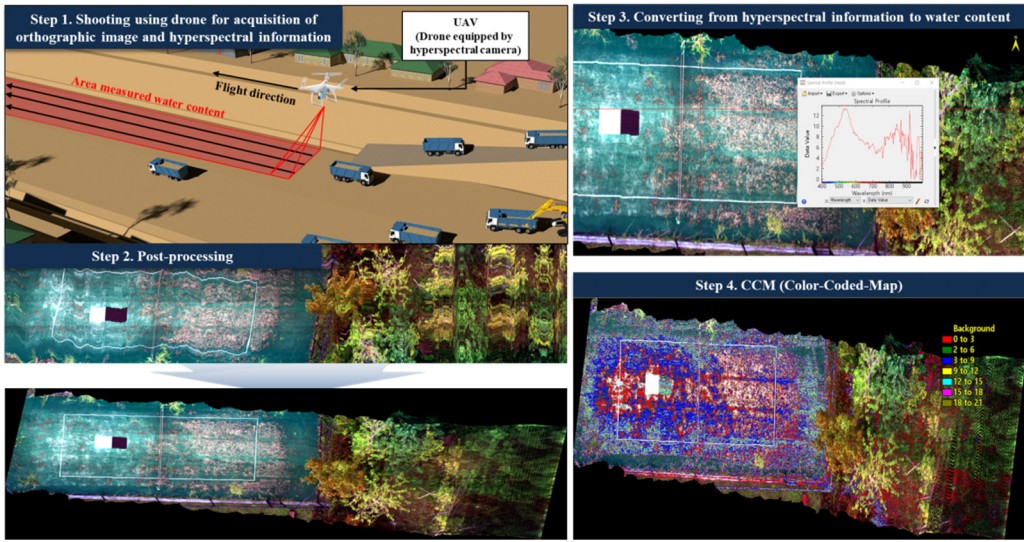

**Figure 1.** Process of color-coded map of water content in the road construction site.

This study aims to present a spectrum index that converts hyperspectral information into groundwater content at the fourth step illustrated in Figure 1. Thus, the spectrum index would refer to the groundwater content, which is a function of reflectance according

to wavelength, as expressed in Equation (1). Here, $w$ refers to water content, $R_i$ refers to reflectance at a wavelength of $i$-nm, and $i$ ranges from 400 nm to 1000 nm. Reflectance ($R$) is the ratio of the reflected energy to the total energy incident on the body, and it is expressed as a percentage. $R$ is expressed through a complex process of reflection, absorption, and transmission of energy; it varies with wavelength and enables features to be identified on the body or surface to be measured [31].

$$\text{Spectrum index} = w = function(R_i) \tag{1}$$

Overall, we aim to present hyperspectral information measured with a hyperspectral camera as a function of Equation (1) (Figure 2), where, $R_i$ in the function may represent one, two, or more points. Corresponding combinations and analyses are described in a later section.

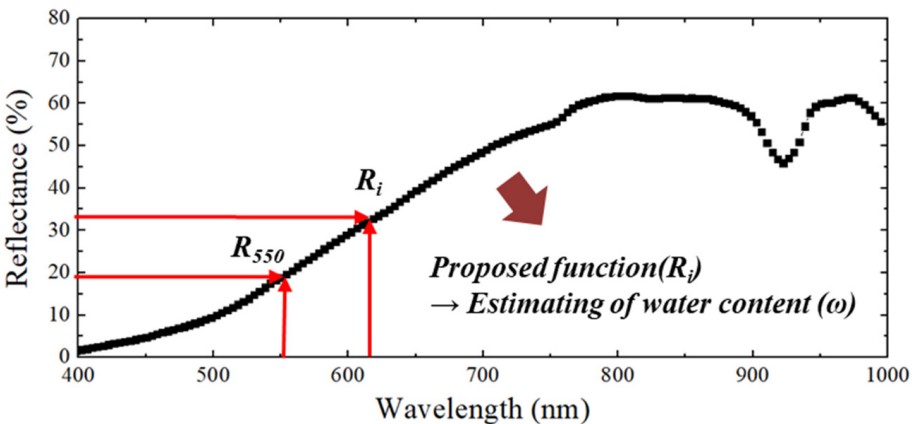

**Figure 2.** Methodology for estimating water content from hyperspectral information.

## 3. Laboratory Tests for Obtaining Hyperspectral Information

### 3.1. System for Obtaining Hyperspectral Information

An indoor experimental system (Figure 3a) was created to acquire hyperspectral information for determining the water content of the soil. The system consists of a hyperspectral camera (Micro HSI410shark, Coring, Seoul, Korea) capable of measuring wavelengths of 400–1000 nm at 2 nm intervals, a stage for push-broom scanning, and software to express the reflectance by wavelength in row data and graphs.

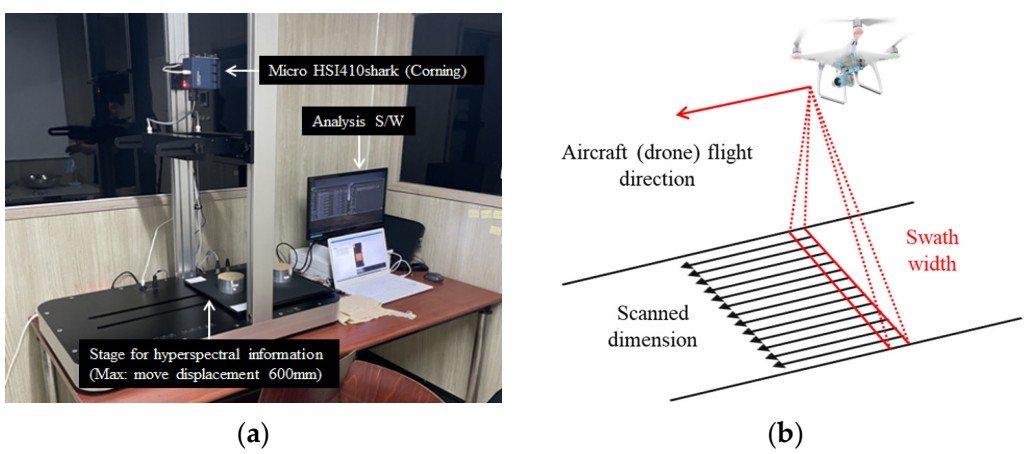

**(a)**          **(b)**

**Figure 3.** System setup: (**a**) laboratory conditions and; (**b**) Push-broom scanning.

The shooting method of a spectral camera is typically divided into staring or spectral scanning, which captures the entire scene in band-sequential format, and push-broom

scanning, which generates a hyperspectral image in a line-by-line format [32]. The push-broom scanning method is a reliable method [33–35] mostly used for aviation photography using an unmanned aerial vehicle. The method also provides reasonable spatial resolution and high spectral resolution [36].

In push-broom scanning, a spectral camera is mounted to have a slit perpendicular to the moving direction of the drone (Figure 3b) to extract spectral information from a designated area of one pixel. The area of one pixel is determined by the sensing interval, focal length, and flight altitude. The drone captures all pixels corresponding to the orthogonal area while moving and then measures the frame of the line corresponding to the next pixel. The captured information is spectral information, including orthographic images.

In the actual field, push-broom scanning is applied as the drone moves, but in the indoor experiment, the movement of the sample located at the bottom was simulated. The simulation was intended to reduce errors due to changes in focus resulting from the camera movements and to omit the geometric correction step performed after actual on-site image acquisition.

### 3.2. Laboratory Test of Soil Sample

The ground measured in this study was a construction site mainly comprising sand composed of soil particles of various sizes, with differences in void ratio, compaction curve, and optimal water content depending on the particle size distribution. As the ground was composed of various sand types, differences were expected in the spectral information measured according to the water content. Therefore, it was necessary to acquire spectral information for various types of ground and convert it into water content. Thus, the derived water content should be constant regardless of the type of ground.

Therefore, for basic normalization, the base soil was set as standard sand. Standard sand is an aggregate used to improve the strength of cement, referring to poorly graded sand, which is granular soil with a particle size of 0.075–2.00 mm in accordance with [37]. Various ground simulations were performed with the addition of 10%, 20%, and 30% fine contents (particle size of 0.075 mm or less).

### 3.2.1. Sieve Test

The sieve test of standard sands with 0, 10, 20, and 30% fine contents was performed according to [38], and the results were as presented in Figure 4 and Table 1. The fine contents were added in relation to the total weight of the soil. As a result of the test, $D_{10}$ was not measured at 20% and 30% of fine contents, but the pass rate for fine contents did not exceed 50%; $D_{10}$ increased as it acted as a denominator in the coefficient of uniformity and the coefficient of curvature. Therefore, all samples used as a result of classification according to [39] could be classified as Poor Sand with uneven particle sizes.

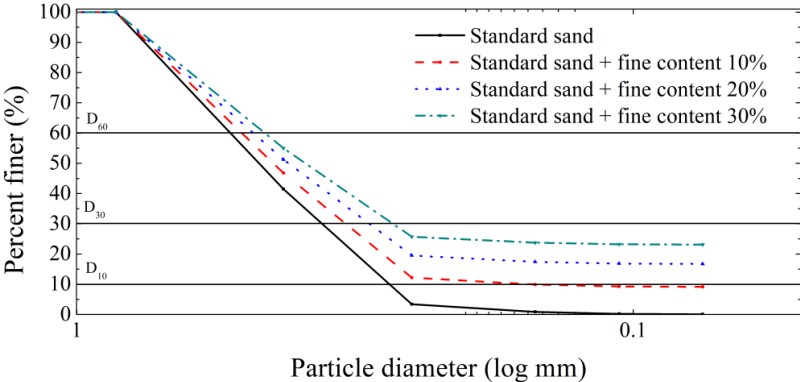

**Figure 4.** Particle size distribution curves of soil samples.

**Table 1.** Sieve analysis results of the used soil sample.

| Fine Content in Standard Sand (%) | $D_{10}$ [1] (mm) | $D_{30}$ [2] (mm) | $D_{60}$ [3] (mm) | Coefficient of Uniformity, $C_u$ [4] | Coefficient of Curvature, $C_c$ [5] | Percentage Passing No. 200 Sieve (%) | Soil Classification |
| --- | --- | --- | --- | --- | --- | --- | --- |
| 0 | 0.274 | 0.363 | 0.530 | 1.934 | 0.907 | 0.06 | SP |
| 10 | 0.150 | 0.329 | 0.505 | 3.367 | 1.429 | 9.15 | SP |
| 20 | - | 0.300 | 0.482 | - | - | 16.72 | SP |
| 30 | - | 0.272 | 0.461 | - | - | 23.13 | SP |

[1] $D_{10}$: Particle diameter in percent finer of the soil corresponding to 10%; [2] $D_{30}$: Particle diameter in percent finer of the soil corresponding to 30%; [3] $D_{60}$: Particle diameter in percent finer of the soil corresponding to 60%; [4] $C_u$: Coefficient of uniformity that calculated by $C_u = D_{60}/D_{10}$; [5] $C_c$: Coefficient of curvature that calculated by $D_{30}^2/(D_{10}D_{60})$.

### 3.2.2. Standard Compaction Test

To produce soil samples with the same degree of compaction, it is necessary for us to know the maximum dry unit weight of each sample. Accordingly, the sieve test of standard sands with 0, 10, 20, and 30% fine contents was performed according to [40], and the results are presented in Figure 5. As the amount of fine content increased, the voids for the volume of the soil sample decreased, being filled with the fine content. Therefore, as shown in Figure 5, the optimum water content increased while the dry unit weight decreased with an increasing amount of fine content.

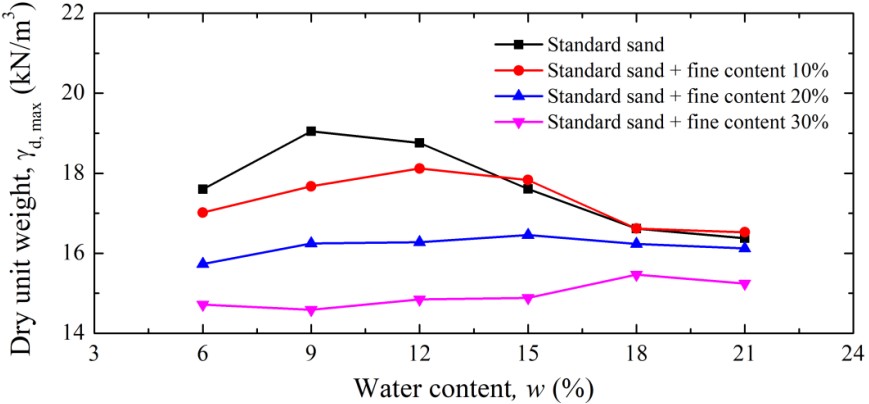

**Figure 5.** Compaction curves of soil samples.

### 3.2.3. Composition of Specimens

In general, the degree of compaction (ratio of on-site dry unit weight and maximum dry unit weight obtained through indoor experiments) is 95% at road construction sites. Thus, the soil sample specimens in this study were prepared with a compaction of 95%. Following the compaction curve in Figure 5, each sample was prepared with water content according to Figure 6a, followed by compaction as shown in Figure 6b. Here, the water content was set to 0–15%, making a total of 16 levels. Finally, specimens (Figure 6c) were obtained and placed on the specimen stage of the configured system (Figure 3a) to extract hyperspectral information.

The water content was the weight ratio of soil to water, and water as much as the water content set in this study was added to the weight of the sample. The volume of the experimental can (diameter = 10 cm, height = 6 cm) used for making the specimens was 471 cm³, and the maximum weight of soil that the experimental can could contain was 914 g (standard sand), 870 g (standard sand + fine content 10%), 790 g (standard sand + fine content 20%), and 742 g (standard sand + fine content 30%), according to the maximum dry unit weights presented in Figure 5.

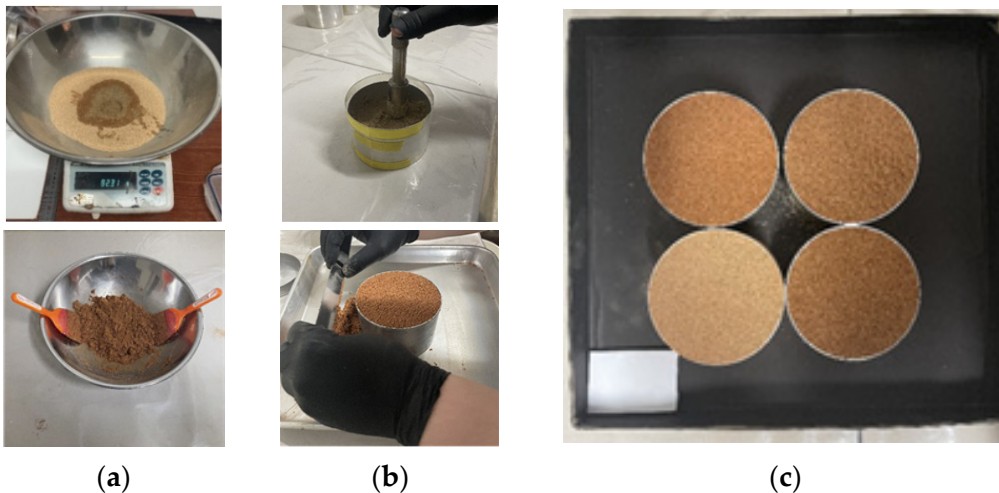

|     |     |     |
| :-: | :-: | :-: |
| (**a**) | (**b**) | (**c**) |

**Figure 6.** Composition of specimens: (**a**) Mixing of a soil sample with standard sand and water; (**b**) Compaction of the specimen in a circular petri; (**c**) Specimens of soil samples with fine and water contents.

Whenever the water content increased by 1%, 9.1 g, 8.7 g, 7.9 g, and 7.4 g of water was added. The composition of specimens was processed precisely, and when the water content was measured again after obtaining the spectral information, the water content was found to be the same as in the initial state. Hence, the measured spectral information reflected the water content of specific specimens accurately.

### 3.3. Hyperspectral Information of Soil Sample

Figure 7 illustrates the hyperspectral information (relationship between wavelength and reflectance) of soil samples according to the water content measured through the system. In all the indoor experimental results, the reflectance according to wavelength showed a similar trend. The reflectance increased non-linearly as the wavelength increased, showing a rapid increase at approximately 750 nm. The maximum reflectance was measured at 800 nm and gradually decreased to the vicinity of the wavelength of 920 nm. Subsequently, the reflectance exhibited non-linear behavior; for example, it increased again. According to the characteristics of each wavelength band, the reflectance increased in the visible-rays region (400–800 nm), and then it decreased and increased non-linearly in the wavelength band of infrared rays (800–1000 nm). In addition, as the water content increased, the reflectance according to the wavelength decreased. This may have been due to absorption occurring more than reflection as the amount of water increased.

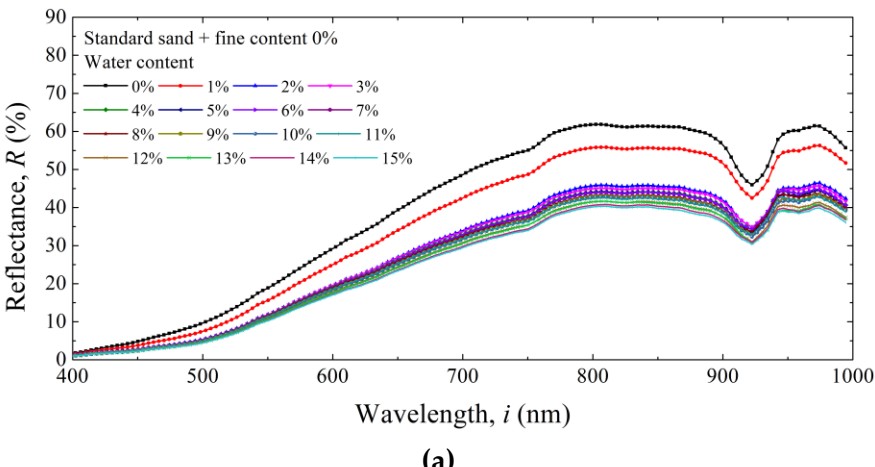

(**a**)

**Figure 7.** *Cont.*

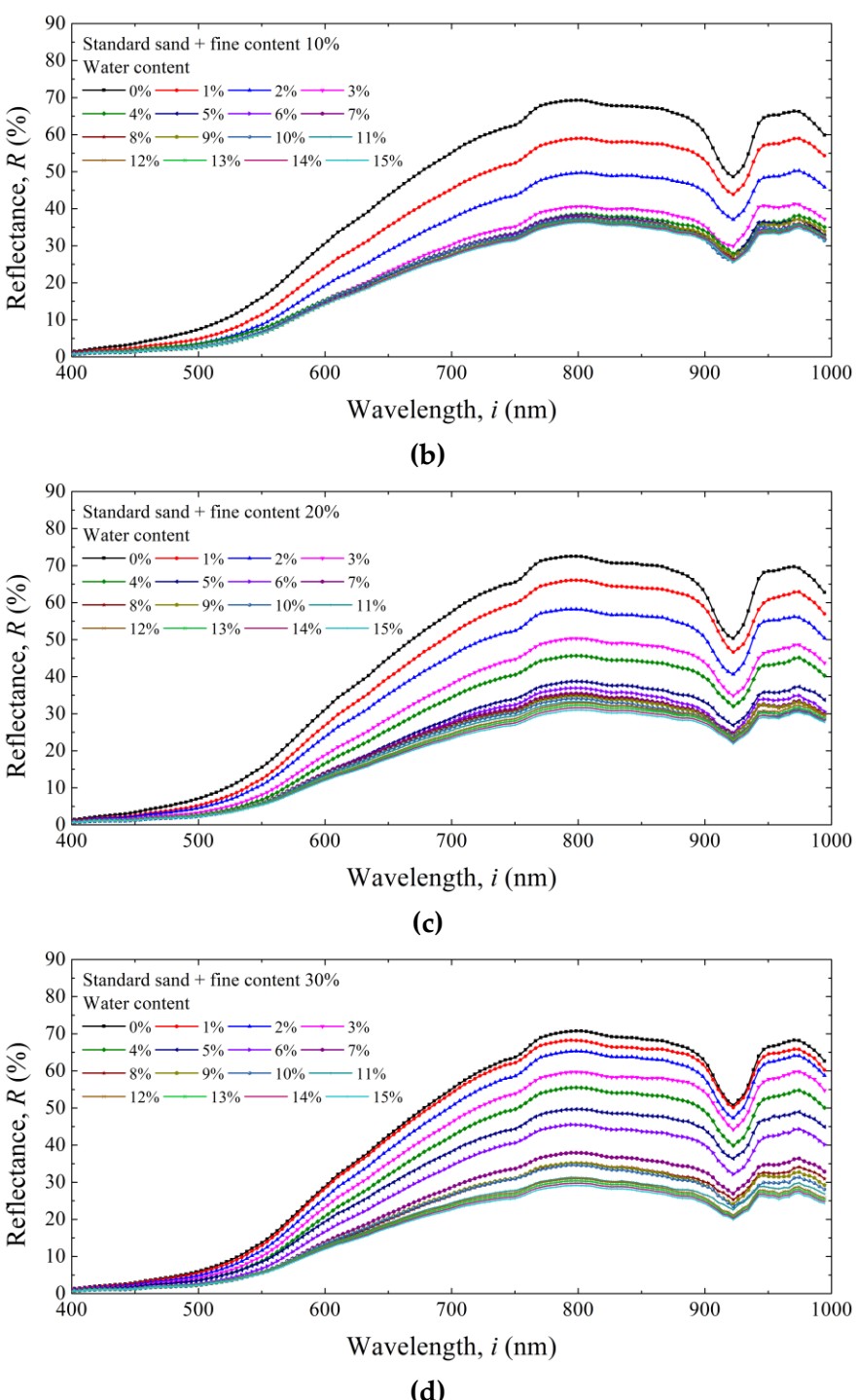

**Figure 7.** Relationship between wavelength and reflectance at fine contents of (**a**) 0%, (**b**) 10%, (**c**) 20%, and (**d**) 30%.

## 4. Estimation of the Spectrum Index for Water Content Prediction

### 4.1. Variability Analysis of Hyperspectral Information

For water content prediction, spectral information measured through experiments should be converted into a single-value spectrum index, which should be inserted into a water content prediction equation. Processing spectral information is time-consuming as numerous row data are collected from the row data obtained with a hyperspectral camera; this phenomenon is due to the relationship between wavelength and reflectance, as shown in Figure 7. However, as water content measurement is performed on the day

of construction in actual road construction sites, with frequent changes made according to various conditions (rainfall, humidity, and temperature conditions), it is essential to minimize the processing time. Accordingly, a significant point (reflectance at a specific wavelength) should be extracted, which should be converted into a spectrum index.

The conditions under which a specific wavelength was selected are illustrated in Figures 8 and 9. The hyperspectral information processed was the ratio of fine content (0–30%, 10% increment) and water content (0–15%, 1% increment), and the wavelength–reflection curves were 64 in total.

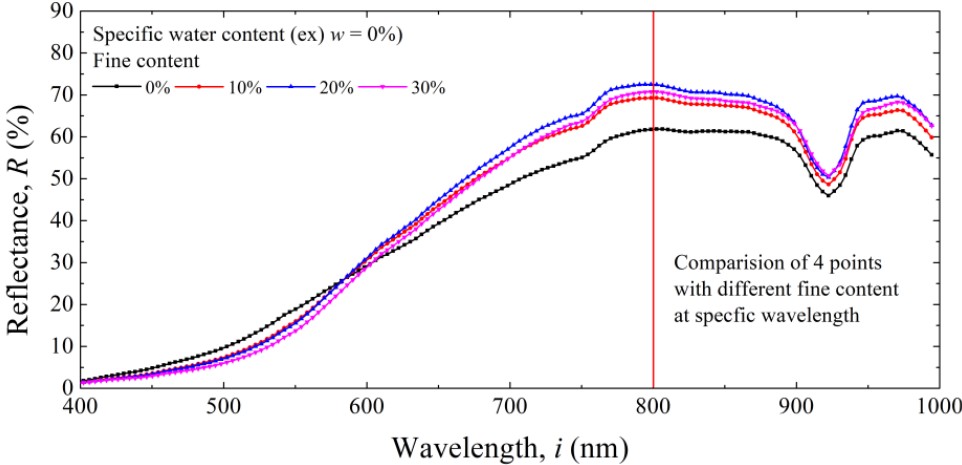

**Figure 8.** Selection of wavelength–reflection point for minimizing fine content effect.

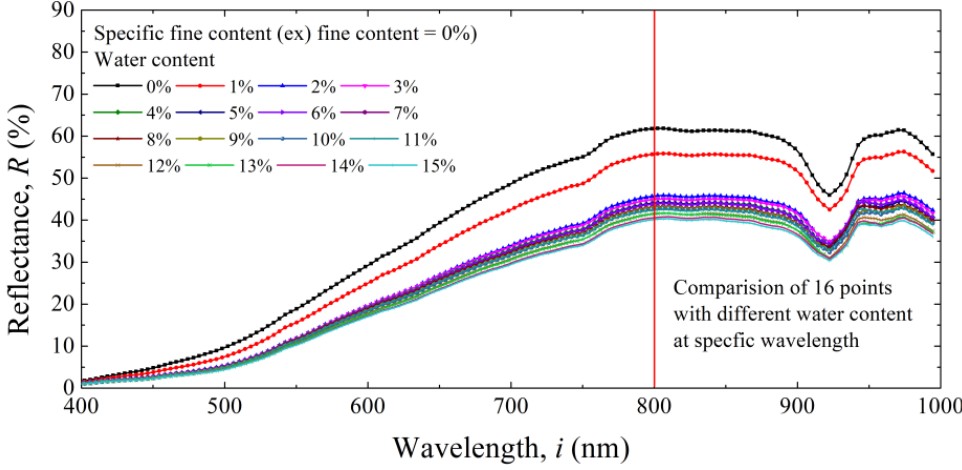

**Figure 9.** Selection of wavelength–reflection point for maximizing water content effect.

The first wavelength selection condition is that the difference in reflectance at a specific wavelength should be small regardless of the content of fine particles at a specific water content amount. If the difference between reflectance is large according to the content of fine particles, a different spectrum index and equation of water content prediction should be selected according to each ground condition. However, this step is practically impossible as it requires setting the ground conditions individually in a large construction site. Therefore, it is necessary to calculate a specific wavelength band with little difference in reflectance according to the change in fine content; this wavelength should show reflectance with slight variability (Figure 8).

The second condition is that there should be a clear difference in reflectance among water contents (Figure 9). If the difference in reflectance at a specific wavelength is not large, there is a possibility that the difference in water content may change rapidly even with a small change in reflectance. Hence, the error would be substantial, having a large impact

on the final product, CCM. Therefore, it is necessary to determine a specific wavelength band with a large difference in reflectance according to a change in water content, which is equivalent to finding a point with a large variability.

Data variability can be evaluated by the coefficient of variation (COV), as in Equation (2). Here, COV is the ratio of standard deviation (SD) to mean. In general, COV is excellent at less than 10%, good from 10% to 20%, acceptable from 20–30%, and not acceptable beyond 30%.

$$COV(\%) = (SD/Mean) \times 100(\%) \tag{2}$$

### 4.1.1. Effects of Fine Contents

Figure 10 presents the COV (Dot in figure) of four points of fine content (0%, 10%, 20%, and 30%) and the average (Red line in figure) of all data according to the wavelength at a specific water content. In terms of average COV, the wavelength with the maximum COV was 29.03% at 500 nm, and the wavelength with the minimum COV was 720 nm, which was 10.23%. In the wavelength range of approximately 400–600 nm, the COV was high; in the range of 600–880 nm, it showed a smooth parabolic shape with a value of 10.23–12.40%. At wavelengths above 880 nm, the COV was approximately steady with slight fluctuations. Regarding the first condition for selecting the wavelength to be applied to the spectrum index, a wavelength with small fine content variability should be selected. For the optimal condition, it is appropriate to use the reflection of a wavelength of 720 nm although it also appears appropriate to use the reflectance of a wavelength of 600 nm or more, as the COV difference from 600 nm to 880 nm was approximately 2%.

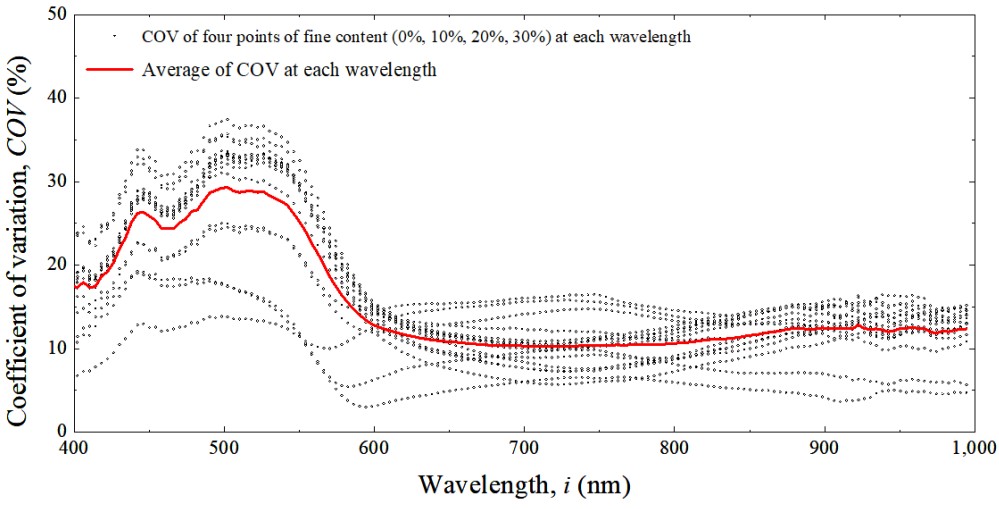

**Figure 10.** Analysis of using COV to select an appropriate wavelength for minimizing fine content effect.

### 4.1.2. Effects of Water Contents

Figure 11 illustrates the COV of reflectance by water content according to fine content amount as well as the average according to wavelength. Regarding the second condition for selecting the wavelength to be applied to the spectrum index, a wavelength with high water content variability should be selected. Therefore, to maximize the water content effect at the wavelength of 600–880 nm, a wavelength with a high COV was selected. Within that range, a minimum COV of 24.31% at 810 nm was measured as well as a maximum COV of 27.60% at 600 nm.

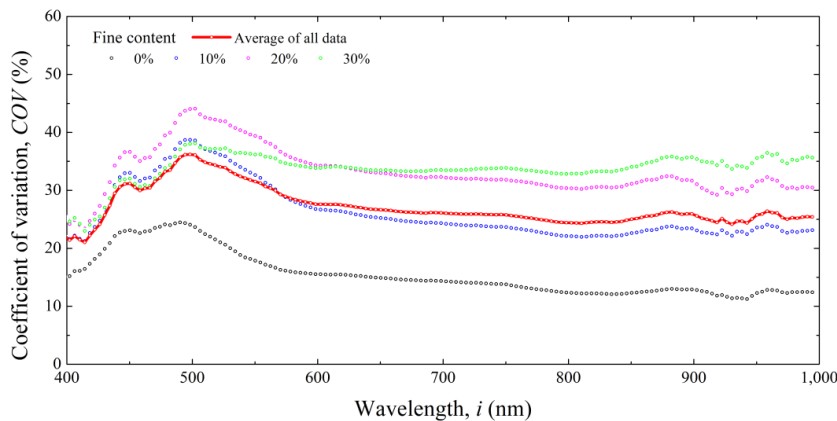

**Figure 11.** Analysis of COV for selecting appropriate wavelength for maximizing water content effect.

### 4.2. Spectrum Index Reflected by Selected Wavelength and Reflection

Following the variability analyses, a specific wavelength for spectrum index was selected as illustrated in Figure 12: (1) 720 nm wavelength and (2) 600–880 nm wavelength. The wavelength of 720 nm is the point with the least fine content effect. As water content had a high variability, it is most appropriate to use the wavelength from a single perspective. In this case, the spectrum index refers to a reflectance at 720 nm. We express the corresponding spectrum index as $R_{720}$. The variability analysis showed that COV exhibited a similar trend in the wavelength band of 600–880 nm. Therefore, all reflectance in the wavelength band of 600–880 nm are considered, and the spectrum index is expressed as an integral. In this paper, the integral is expressed as $I_{600-820}$ (integral from 600 nm to 820 nm of wavelength), and it refers to the area between the wavelength–reflectance curve and the *x*-axis (range of 600–820 nm).

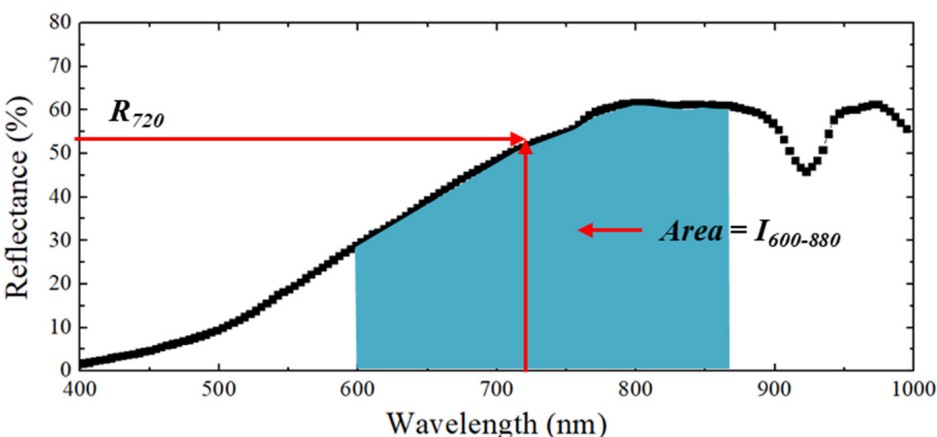

**Figure 12.** Selection of wavelength and reflection for spectrum index.

### 4.3. Equation for Predicting Water Content Using Spectrum Index

Data fitting was performed, as shown in Figure 13, to determine the water content prediction formula using the appropriate spectral index. The fine particle content was not classified separately in all the data here; $R_{720}$ and $I_{600-880}$ were plotted on the *x*-axis against water content on the *y*-axis. The total number of data was 64. Figure 13 shows that the water content gradually decreased as the spectrum index increased, but the relationship was nonlinear. Therefore, it is necessary to derive a non-linear equation for the relationship.

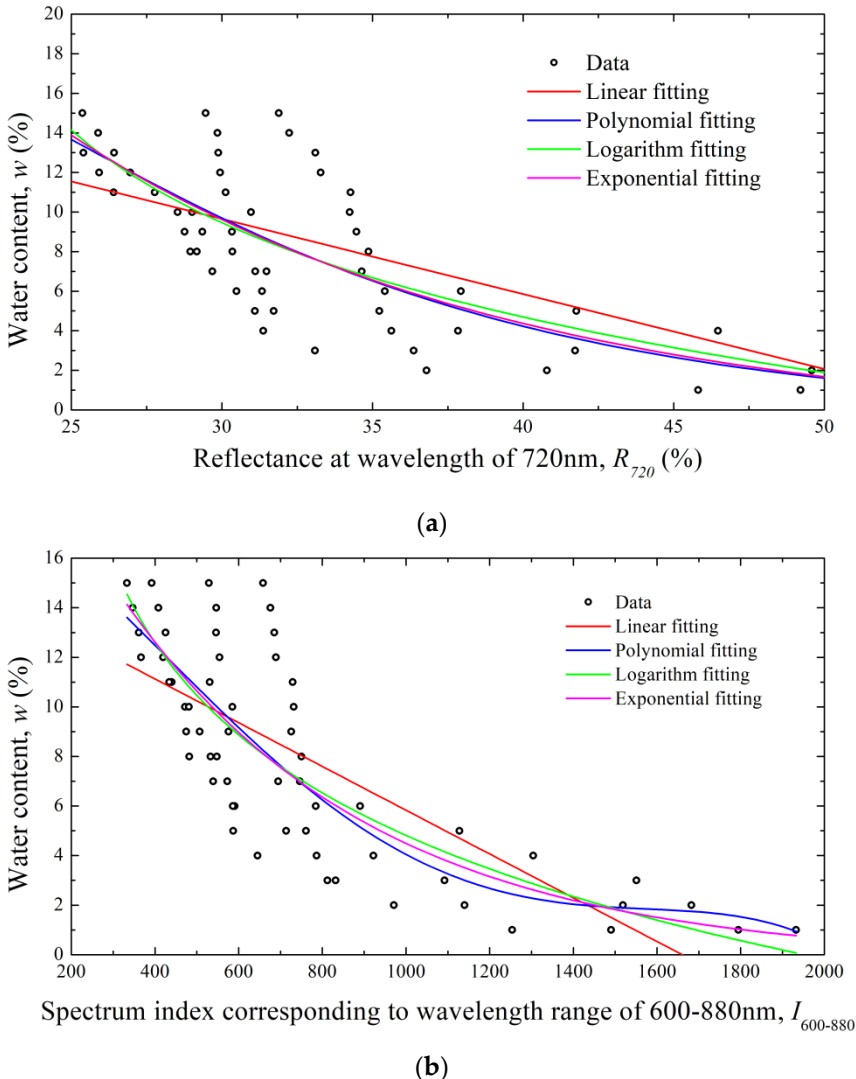

**Figure 13.** Spectrum index with water contents; (**a**) $R_{720}$ and (**b**) $I_{600-820}$.

Various fitting equations, including linear, polynomial, logarithmic, and exponential equations, were considered. Table 2 presents the equations and their corresponding $R^2$ values. After fitting, $R^2$ was low for $I_{600-880}$ (using the integral area) compared with that of $R_{720}$ (calculated as a single point). Therefore, it was appropriate to select $R_{720}$ as a spectrum index; an exponential fitting model with a high correlation coefficient was selected as the equation for water content prediction.

**Table 2.** Results of the fitting.

| Index | Fitting Model | Equation | $R^2$ |
|---|---|---|---|
| $R_{720}$ | Linear | $w = -0.379R_{720} + 21.021$ | 0.636 |
| | Polynomial | $w = -8.38 \times 10^{-6}R_{720}{}^4 + 0.0012R_{720}{}^3 - 0.0462R_{720}{}^2 - 0.2631R_{720}122(0 \text{ index} + 33.7973)$ | 0.687 |
| | Logarithm | $w = 25.767 - 7.004\ln(R_{720} - 19.738)$ | 0.695 |
| | Exponential | $w = -1.172 + 79.648\exp(-0.0666R_{720})$ | 0.697 |
| $I_{600-880}$ | Linear | $w = -0.00883I_{600-880} + 14.658$ | 0.579 |
| | Polynomial | $w = -5.37 \times 10^{-12}I_{600-880}{}^4 + 2.159 \times 10^{-8}I_{600-880}{}^3 - 7.782 \times 10^{-5}I_{600-880}{}^2 + 0.0479I_{600-880}122(0 \text{ index} + 10.0868)$ | 0.637 |
| | Logarithm | $w = 48.239 - 6.432\ln(I_{600-880} - 144.708)$ | 0.643 |
| | Exponential | $w = -0.2228 + 25.032\exp(-0.00167I_{600-880})$ | 0.645 |

### 4.4. Comparison of the Literature with Proposed Spectrum Index Method

To verify the suitability of the water content prediction equation proposed in this study, a comparison with the existing theoretical equations was performed. Eleven of the thirty high-$R^2$ prediction equations investigated by Ge et al. [30] are presented in Table 3. Because the existing equations target only the spectrum index, a separate fitting should be performed for the water content prediction equation. According to [30], a linear fitting was performed. Therefore, to obtain the equation for predicting water content, the spectral information from this study was substituted into the spectrum index, and the equations were obtained individually through linear fitting.

**Table 3.** Comparison of bias factors of each inflection point.

| Spectrum Index | | Equation for Water Content Prediction | Ref. |
|---|---|---|---|
| Sort | Equation | | |
| mNDVI705 | $(R_{750} - R_{705})/(R_{740} + R_{705} + 2R_{445})$ | $w = -105.01 \text{mNDVI705} + 13.40$ | [41] |
| NDVI | $(R_{800} - R_{680})/(R_{800} + R_{680})$ | $w = 161.11 \text{NDVI} - 20.57$ | [42] |
| NDCI | $(R_{762} - R_{527})/(R_{762} + R_{527})$ | $w = 8.04 \text{NDCI} + 1.44$ | [41] |
| NDVI705 | $(R_{750} - R_{705})/(R_{750} + R_{705})$ | $w = -120.54 \text{NDV705I} + 14.53$ | [43] |
| RVI | $R_{800}/R_{680}$ | $\omega = 55.28 \text{RVI} - 71.13$ | [43] |
| NDRE | $(R_{750} - R_{705})/(R_{750} + R_{705})$ | $w = -120.54 \text{NDRE} + 14.53$ | [44] |
| GNDVI | $(R_{750} - R_{550})/(R_{750} + R_{550})$ | $w = 3.79 \text{GNDVI} + 5.10$ | [45] |
| OSAVI | $[(1 + 0.16)(R_{800} - R_{670})]/(R_{800} + R_{670} + 0.16)$ | $w = 115.04 \text{OSAVI} - 19.52$ | [42] |
| VOG1 | $R_{740}/R_{720}$ | $w = -114.33 \text{VOG1} + 127.58$ | [46] |
| VOG2 | $(R_{734} - R_{747})/(R_{715} - R_{726})$ | $w = 2.71 \text{VOG2} + 5.66$ | [46] |
| VOG3 | $(R_{734} - R_{747})/(R_{715} + R_{720})$ | $w = 577.11 \text{VOG3} + 14.58$ | [46] |

Figure 14 illustrates the results of the water content prediction equation proposed in this study and the measured and predicted water contents according to the existing theoretical formulas. Overall, in the existing equations, no significant change was observed in the predicted water content as the measured water content increased. In other words, the $R^2$ value was distributed from 0.002 to 0.122, indicating an extremely low correlation. Therefore, when the spectral information obtained in this study was substituted into the existing spectrum index, a considerable error was obtained, demonstrating that the water content prediction equation using the proposed $R_{720}$ is appropriate.

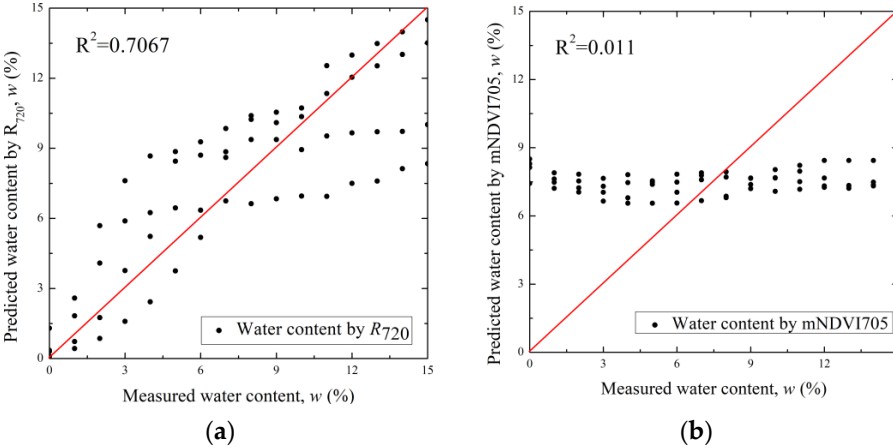

**Figure 14.** *Cont.*

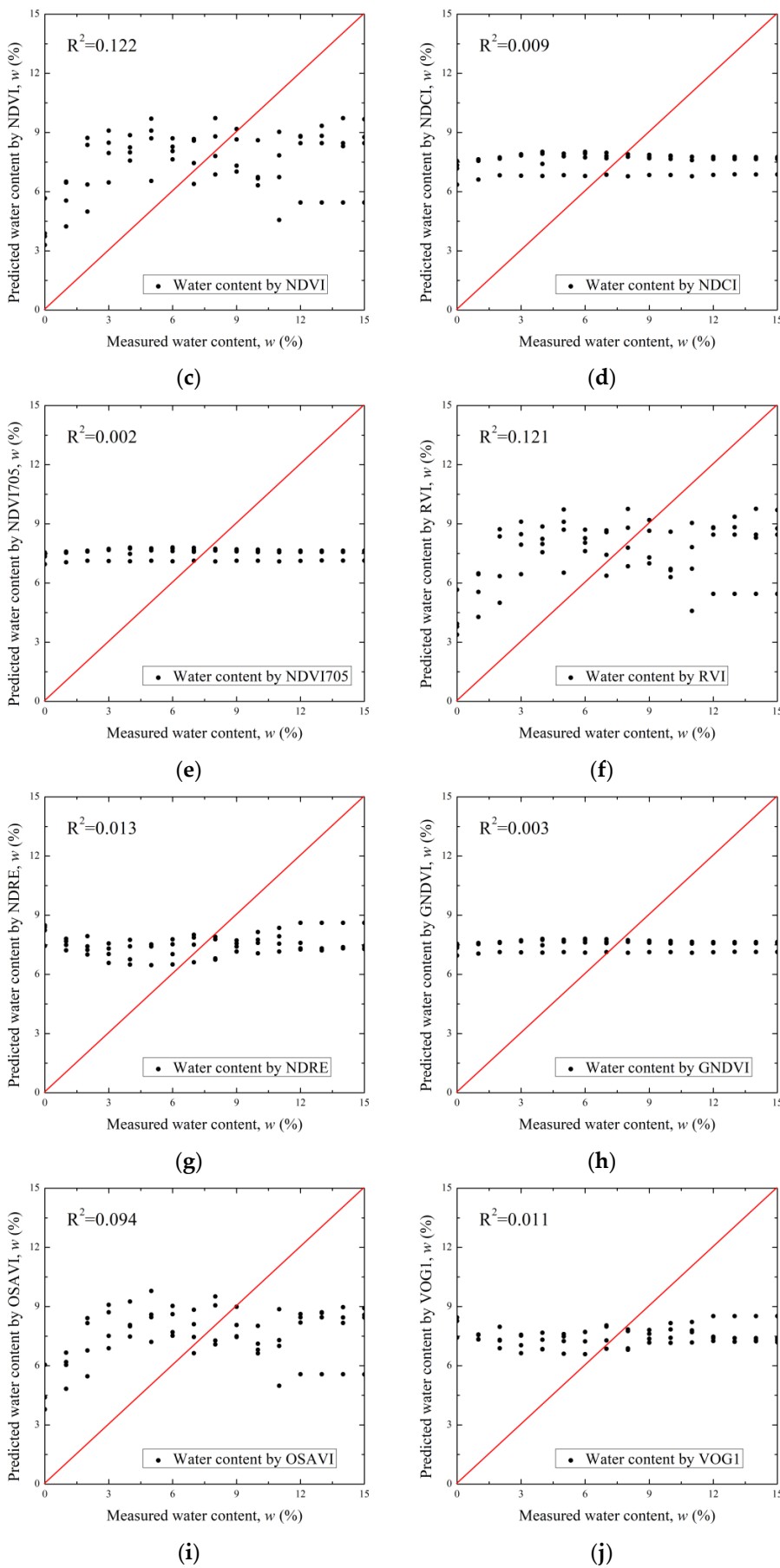

**Figure 14.** *Cont.*

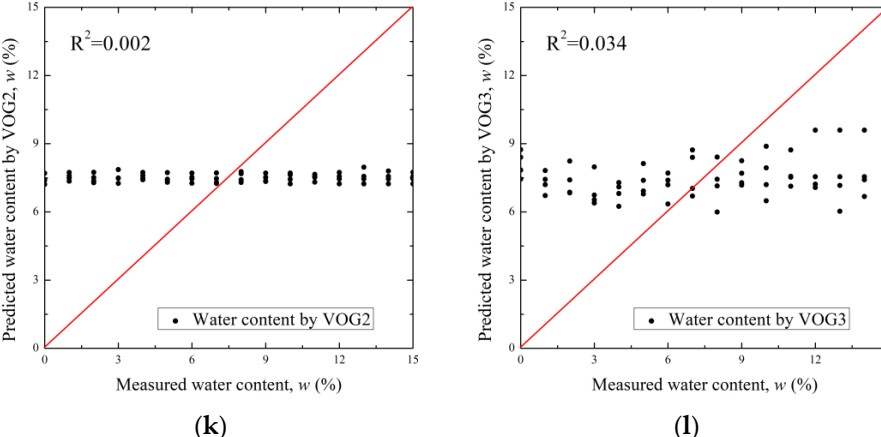

**Figure 14.** Comparison of predicted and measured water contents by (**a**) $R_{720}$, (**b**) mNDVI705, (**c**) NDVI, (**d**) NDCI, (**e**) NDVI705, (**f**) RVI, (**g**) NDRE, (**h**) GNDVI, (**i**) OSAVI, (**j**) VOG1, (**k**) VOG2, and (**l**) VOG3.

## 5. Conclusions

In this study, groundwater content was measured for determining road bearing capacity and for road quality control during road earthworks and pavement construction toward achieving "sustainable roads." The existing water content measurement method is the point measurement method; however, we aimed to acquire the water content of a wide area at once, necessitating the use of hyperspectral information. Hence, hyperspectral information was obtained through many indoor experiments, and a water content prediction formula was proposed. The conclusions drawn from this study are as follows.

1. In this study, sophisticated specimens were created by adding fine contents to standard sand, and hyperspectral information was obtained according to water content through precise laboratory tests. For hyperspectral information, a spectrum index was selected through various correlation analyses, and an equation to convert the spectrum index to water content was proposed.

2. The suitable wavelength for calculating the spectrum index was 600–880 nm, as determined through variability analysis based on the water content and fine contents. The variability analysis results showed that no difference existed in the results of the equation for water content prediction even when a single wavelength within the range was selected. When the integral value of reflectance was used at 600–880 nm, $R^2$ was rather low. This phenomenon was the result of the overlapping variability of wavelength and reflectance. Even when the $R^2$ of the corresponding index was measured, it was not appropriate as it increased the time for calculating the spectrum index.

3. The available equation for the prediction of the groundwater content is when the reflectance at a wavelength of 720 nm is applied to the exponential model. As a result of the linear regression analysis according to the measured and predicted water content, $R^2$ was measured to be the highest, which means that it is most suitable for representing the water content in the ground. In terms of spectral range, 720 nm is deep red light.

4. The correlation ($R^2$: 0.009–0.122) when the existing spectrum index for water content prediction was substituted into the hyperspectral information obtained in this study was measured to be very low. Even when the existing equation was substituted into the hyperspectral information obtained by Ge et al. [26], the $R^2$ ranged from 0.052 to 0.398, indicating that the reliability of the existing formula was low. Therefore, the $R^2$ (0.7067) of our proposed equation for water content prediction according to $R_{720}$ was large and reliable. This is because the existing method calculated the water content in a linear line through a simple linear regression analysis of the spectrum index.

5. The disadvantage of this study is that the proposed equation was derived without going through an actual field test. Thus, in the field, errors may occur depending

on actual variables, such as weather, temperature, humidity, and the skill level of the drone operator. Therefore, it is necessary to test the accuracy and reliability of the equation derived from this study in the field, and the equation must be modified through additional data acquisition.

**Author Contributions:** Conceptualization, K.L., K.S.K., J.P. and G.H.; Data curation, K.L., J.P. and G.H.; Funding acquisition, K.S.K.; Formal analysis, K.L., K.S.K. and J.P.; Investigation, K.L., K.S.K., J.P. and G.H.; Methodology, G.H.; Project administration, J.P. and G.H.; Resources, K.S.K. and G.H.; Supervision, J.P.; Validation, K.L., J.P. and G.H.; Visualization, K.L.; Writing—original draft preparation, K.L. and K.S.K.; Writing—review and editing, K.S.K., J.P. and G.H. All authors have read and agreed to the published version of the manuscript.

**Funding:** This research was funded by the Ministry of Land, Korea Agency for Infrastructure Technology Advancement (KAIA), grant number 22SMIP-A157182-03.

**Institutional Review Board Statement:** Not applicable.

**Informed Consent Statement:** Not applicable.

**Data Availability Statement:** Not applicable.

**Conflicts of Interest:** The authors declare no conflict of interest.

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
