# Peer review of "Spectrum Index for Estimating Ground Water Content Using Hyperspectral Information"

_sustainability, doi:10.3390/su142114318_

Round 1
Reviewer 1 Report
This study aims to use hyperspectral information to estimate the ground water content. This topic is interesting. However, before recommending publication, the following issues need to be revised.
(1) The quality of diagrams needs to improve. For example, Figure 1 is unclear; the signs in Figure 10 cannot be distinguished.
(2) The original data of the camera and the processing procedure of the hyperspectral information need to be provided and explained.
(3) There is a potential drawback of the method. Since the drone can only scan the surface of road, it may not effectively reflect the real water content of the road. What do the authors think about this issue?
(4) To complete the background of this study, the work named “Effect of in-situ water content variation on the spatial variation of strength of deep cement-mixed clay” is recommended to cite.
Author Response
수정사항은 파일로 첨부하였으며, 우리는 당신의 심사에 감사드립니다.
Corrections are attached as files, and we appreciate your review.

Reviewer 2 Report
Places to improve:
-Introduction needs to have the latest development in the area; revise
-Nice graphical presentation though needs proper justifications and explanations; revise the discussions to further connect to the results
--Reference list needs to be updated to cover majority the relevant research specially 2019 onward papers
Author Response
Corrections are attached as files, and we appreciate your review.

Reviewer 3 Report
In this paper, the remote sensing technology of high spectral information is used to effectively measure the content of groundwater in a large area. Through laboratory tests, a prediction equation of water content is established. The idea is innovative, but it lacks the combination with engineering and is recommended to be modified.
1 As we all know, the lithology of the overlying strata of groundwater is variable. Whether the addition of fine sand in the standard sand can meet the simulation conditions, please answer.
2 According to the author 's description of Figure 9, ' The second condition is that there should be a clear difference in reflectance among water contents ( Figure 9 ). ' However, by observing Figure 9, it is found that some curves are approximately coincident. Is this description appropriate?
3 Professional staff are required to modify the issue of article statements.
4 Avoid using some colloquial words and phrases, because they have a negative impact on the mood of writing.
Author Response

(The authors gave the same response as above.)
